# KEY PROTECTED CLASSIFICATION FOR GAN ATTACK RESILIENT COLLABORATIVE LEARNING

## ABSTRACT

Large-scale publicly available datasets play a fundamental role in training deep learning models. However, large-scale datasets are difficult to collect in problems that involve processing of sensitive information. Collaborative learning techniques provide a privacy-preserving solution in such cases, by enabling training over a number of private datasets that are not shared by their owners. Existing collaborative learning techniques, combined with differential privacy, are shown to be resilient against a passive adversary which tries to infer the training data only from the model parameters. However, recently, it has been shown that the existing collaborative learning techniques are vulnerable to an active adversary that runs a GAN attack during the learning phase. In this work, we propose a novel key-based collaborative learning technique that is resilient against such GAN attacks. For this purpose, we present a collaborative learning formulation in which class scores are protected by class-specific keys, and therefore, prevents a GAN attack. We also show that very high dimensional class-specific keys can be utilized to improve robustness against attacks, without increasing the model complexity. Our experimental results on two popular datasets, MNIST and AT&T Olivetti Faces, demonstrate the effectiveness of the proposed technique against the GAN attack. To the best of our knowledge, the proposed approach is the first collaborative learning formulation that effectively tackles an active adversary, and, unlike model corruption or differential privacy formulations, our approach does not inherently feature a trade-off between model accuracy and data privacy.

## 1 INTRODUCTION

Deep neural networks have shown remarkable performance in numerous domains, including computer vision, speech recognition, language processing, and many more. Most deep learning approaches rely on training over large-scale datasets and computational resources that makes the utilization of such datasets possible.

While large-scale public datasets, such as ImageNet (Deng et al., 2009), Celeb-1M (Guo et al., 2016), and YouTube-8M (Abu-El-Haija et al., 2016), have a fundamental role in deep learning research, it is typically difficult to collect a large-scale dataset for problems that involve processing of sensitive information. For instance, data privacy becomes a significant concern if one considers training over all the voice, iris, finger-print, or face samples collected by the users of a mobile application.

To enable training over large-scale datasets without compromising data privacy, a collaborative learning framework (CLF) is proposed by Shokri & Shmatikov (2015). In CLF, the model is trained in a distributed way, where each *participant* contributes to training without sharing its (sensitive) data with other participants. More specifically, each participant hosts only its own training examples, and a central server, called the *parameter server*, combines local model updates into a shared model. Therefore, the training procedure effectively utilizes the data owned by all participants. At the end, the final model parameters are shared with all participants.

However, there are cases where the original CLF approach fails to preserve data privacy due to knowledge embedded in the final model parameters. In particular, Fredrikson et al. (2015) show that the parameters of a neural network model trained on a dataset can be exploited to partially reconstruct the training examples in that dataset. To mitigate this threat, one may consider partially

corrupting the model parameters by adding noise into the final model as formulated by Chaudhuri et al. (2011). However, such prevention mechanisms introduce a difficult trade-off between classifier accuracy versus data privacy level for collaborative training.

The study by Shokri & Shmatikov (2015) also shows that differential privacy (Dwork, 2011) can be incorporated into CLF in a way that guarantees the indistinguishability of the participant data. Following this study, several other differential privacy based approaches, which provide noise injection methods (Phan et al., 2016; Abadi et al., 2016) or training frameworks (Papernot et al., 2016), have recently been proposed.

However, it has recently been shown that CLF can be vulnerable to not only passive attacks, but also much more powerful *active attacks*, *i.e.*, training-time attacks (Hitaj et al., 2017). More specifically, a training participant can construct a generative adversarial network (GAN) (Goodfellow et al., 2014) such that its GAN model learns to reconstruct training examples of one of the other participants over the training iterations. For this purpose, attacker defines a new class for the joint model, which acts as the GAN discriminator, and utilizes the samples generated by its GAN generator when locally updating the model. In this manner, the attacker effectively forces the victim to release more information about its samples, as the victim tries to differentiate its own data from attacker class during its local model updates.

In this paper, we propose a novel collaborative learning formulation that prevents the GAN attack. First, we observe that GAN attack depends on the classification scores of the targeted classes. Based on this observation, we define a classification model where class scores are protected by class-specific keys, which we call *class keys*. To make the CLF approach secure even in the case of a compromised parameter server, our approach generates class keys independently within each training participant and keeps the keys private throughout the training procedure. In this manner, we prevent the access of the adversary to the target classes, and therefore the GAN attack. We also demonstrate that the dimensionality of the keys directly affect the security of the proposed model, much like the length of the passwords. We observe, however, that naively increasing the key dimensionality can greatly increase the number of model parameters, and therefore, reduce data efficiency and classification accuracy. We demonstrate that this issue can be addressed by introducing fixed neural network components that allow using much higher dimensional keys without increasing the model complexity. We experimentally validate that our approach prevents a GAN attack while providing effective collaborative learning on the MNIST (LeCun et al.) and Olivetti Faces (Samaria & Harter, 1994) datasets, both of which are challenging datasets in the context of privacy attacks due to their relative simplicity, and therefore, the ease of reconstructing the samples in them.

The rest of the paper is organized as follows. In Section 2, we provide an overview of the recent research in privacy-preserving machine learning. In Section 3, we present a detailed and technical summary of collaborative learning, GANs, and the GAN attack technique. In Section 4, we discuss the details of our approach. In Section 5, we provide an experimental validation of our model. Finally, in Section 6, we conclude the paper.

## 2 RELATED WORK

Privacy preserving machine learning methods have become popular over the recent years. In this section we describe the most relevant ones to our work.

Differential privacy (DP) defined in Dwork (2011) provides sample indistinguishability, therefore protects privacy of subject data. This concept is first formalized for machine learning in general in Chaudhuri et al. (2011). Recent studies utilize DP in large-scale learning as (i) structured noise addition as in (Abadi et al., 2016; Phan et al., 2016), or (ii) 2-step training methodology as in Papernot et al. (2016).

Collaborative learning framework (CLF, Shokri & Shmatikov (2015)) enables large-scale learning in scenarios where there is no public dataset available. Besides, authors show that DP can be integrated into CLF in such a way that pooled gradients do not leak much information about participants. This leakage can be controlled by the noise parameter $\epsilon$ resulting in a trade-off between utility and accuracy.

Very recently, Hitaj et al. (2017) devised a powerful attack against CLF. In this attack, one of the participants in the CLF is assumed to be the adversary. The adversary tries to exploit one of classes belonging to other participants (victim) by using a generative adversarial network (GAN) (Goodfellow et al. (2014)). This attack is powerful in a sense that adversary can actively influence the victim to release more details about its samples during the training process. Therefore, it is denoted as an active attack, different from the passive one investigated in Fredrikson et al. (2015). To the best of our knowledge no solution has previously been proposed against the GAN attack. In this work, we propose a novel collaborative learning model that is effectively resilient against the GAN attack, and, we do not rely on a model corruption or differential privacy technique to preserve data privacy.

## 3 BACKGROUND

Shokri & Shmatikov (2015) and Hitaj et al. (2017) Our work builds on the collaborative learning framework (Shokri & Shmatikov, 2015) and tackles the generative adversarial network attack problem (Hitaj et al., 2017). Therefore, before providing the details of our approach, we first provide brief background on CLF, GAN, and the GAN attack in the following.

### 3.1 COLLABORATIVE LEARNING OF PARTICIPANTS

The goal of CLF is to collaboratively train a shared model over the private datasets of several participants such that the model generalizes across the datasets of all the participants. For this purpose, CLF defines a protocol where each participant shares information only about its learning progress, rather than the data directly, with the others over the training iterations. Locally, participants train their model as usual using gradient based optimization, but share with others fractions of changes in model parameters, at predefined intervals. The framework is set up among participants based on the following components and the associated policies:

(i) A mechanism for participants to share parameter updates. This is typically realized by a trusted third-part parameter server (PS). As its name suggests, it is a platform where participants accumulate their model updates by means of uploading or downloading a predefined fraction of parameter changes during training. It is also possible to follow a gradient sharing scheme.

(ii) A common objective and model architecture. All participants use the same model architecture and training objective. Typically, participants declare class labels for which they have training data.

(iii) Meta-parameters. The hyper-parameters of the CLF setup, such as the parameter download fraction ($\theta_d$) and the upload fraction ($\theta_u$), gradient clipping threshold ($\gamma$), and the order of participants during training (e.g., round robin, random, or asynchronous) are typically predetermined. See Shokri & Shmatikov (2015) for a full list of meta-parameters.

Once the framework is established, participants start training on their local datasets in a predetermined order. When a participant takes turn, it first downloads $\theta_d$ fraction of parameters from the PS and replaces them with its local parameters. After performing one epoch of training on its local dataset, participant uploads $\theta_u$ fraction of resulting gradients to the PS. It is also possible to incorporate differential privacy to guarantee a level of sample indistinguishability for enhanced privacy protection.

### 3.2 GENERATIVE ADVERSARIAL NETWORK

Generative adversarial network (Goodfellow et al., 2014) is an unsupervised learning process for learning a model of the underlying distribution of a sample set. A GAN model consists of two sub-models, called generator and discriminator. The generator corresponds to a function $G(z; \theta_G)$ that aims to map each data point $z$ sampled from a prior distribution, *e.g.* uniform distribution $\mathcal{U}(-1, 1)$, to a point in the data space, where $\theta_G$ represents the generator model parameters. Similarly, the discriminator is a function $D(x; \theta_D)$ that estimates the probability that a given $x$ is a real sample from the data distribution, where $\theta_D$ represents the discriminator model parameters.

The generator and discriminator models are trained in turns, by playing a two-player minimax game. At each turn, the generator is updated towards generating samples that are indistinguishable from

the real samples according to the current discriminator's estimation:

$$\min_{\theta_G} \ \mathbf{E}_z\big[\log(1 - D(G(z; \theta_G)); \theta_D)\big]. \tag{1}$$

The discriminator, on the other hand, is updated towards distinguishing the samples given by $G$ from the real ones:

$$\max_{\theta_D} \ \mathbf{E}_x\big[\log(D(x; \theta_D)\big] + \mathbf{E}_z\big[\log(1 - D(G(z; \theta_G)); \theta_D)\big]. \tag{2}$$

GANs have successfully been utilized in numerous problems, *e.g.* see Luc et al. (2016); Nguyen et al. (2016); Zhang et al. (2017); Yeh et al. (2017).

### 3.3 GAN Attack in Collaborative Learning Framework

Hitaj et al. (2017) devise a powerful GAN-based active attack for the collaborative learning framework. In this scenario, an adversarial participant takes places during training in a CLF setup, (incorrectly) declares that it hosts some class $c_{\text{fake}}$ and tries to extract information about some class $c_{\text{attack}}$[1] Hitaj et al. (2017) show that the adversary can execute an attack by secretly training a generator model for the class $c_{\text{attack}}$, and treating the shared model as a discriminator. Such an approach effectively turns CLF into a GAN training setup where the adversary takes the following steps:

(i) Adversary updates its generator network towards producing samples that are classified as class $c_{\text{attack}}$, by the shared classification model, as in Eq. (1).
(ii) Adversary takes samples from its generator, labels them as $c_{\text{fake}}$ and updates the shared classification model towards classifying the synthetic samples as class $c_{\text{fake}}$, as in Eq. (2).

The GAN attack works in two ways. First, throughout the training iterations, the adversary continuously updates its generator, therefore, it can progressively improve its generative model and the reconstructions that it provides. Second, since the adversary defines the class $c_{\text{fake}}$ as part of the shared model, the participant that hosts $c_{\text{attack}}$ updates the shared model towards minimizing the misclassification of its training examples into class $c_{\text{fake}}$. Over the iterations, this practically forces the victim into releasing more detailed information about the class $c_{\text{attack}}$ while updating the shared model (Hitaj et al., 2017). This latter step makes the GAN attack particularly powerful as it influences the training of all participants, and, it is also the main reason why the technique is considered as an active attack.

## 4 Proposed method

In this section, we describe the details of our proposed approach to prevent the GAN attack. Our model relies on the distributed learning framework proposed by Shokri & Shmatikov (2015) and we assume that there is an active adversary as in Hitaj et al. (2017). We explain the details of our key-protected classification model in Section 4.1. Then, we propose an extension of our approach that enables efficient incorporation of high dimensional keys, which reduces the success likelihood of a GAN attack, in Section 4.2.

### 4.1 Key Protected Classification Model

Our starting point is the observation that GAN attack relies on the knowledge of classification score of the target class throughout the training iteration, as also discussed in Section 3.3. To prevent a GAN attack in a collaborative learning setup, we aim to mathematically prevent each participant from estimating the classification scores for the classes hosted by the other participants. For this purpose, we introduce class-specific keys for all classes and parametrize the classification function in terms of these keys in a way that makes classification score estimation without keys practically improbable.

For this purpose, we require each participant to generate a random class key for its classes during initialization, and keep it private until the end of the training procedure. We denote the key for

---

[1]The adversary may additionally have its own real classes and a real dataset, but for the sake of simplicity, we assume that the adversary works only on its privacy attack.

class $c$ by $\psi(c) \in \mathbb{R}^{d_k}$, where $d_k$ is the predetermined dimensionality of each key. For the sake of simplicity, we assume that there is a single adversary, and participants do not have overlapping classes in our main experiments. However, our formulation naturally extends to multiple attackers. In addition, later in Section 5.5, we relax our latter assumption and show how our formulation can behave well when participants have shared classes, yet private class keys.

Our goal is to train a deep (convolutional) neural network for classification. However, unlike the traditional neural networks that directly output classification scores, we define the network with model parameters $\theta$ as an embedding function $\phi_\theta(x)$ that maps each given input $x$ (*e.g.*, an image) from the source domain to a $d_k$-dimensional vector. Then, we define the classification score for class $c$ by a simple dot product between the embedding output and the class key:

$$\langle \psi(c), \phi_\theta(x) \rangle, \tag{3}$$

where $\psi(c), \phi_\theta(x) \in \mathbb{R}^{d_k}$. Therefore, the final classification takes the form of choosing the class whose key leads to the maximum classification score:

$$\arg \max_{c \in C^{\text{all}}} \langle \psi(c), \phi_\theta(x) \rangle, \tag{4}$$

where $C^{\text{all}}$ is collection of all classes among all participants. We assume that all class keys and also the output of the embedding network are $\ell_2$-normalized[2], for two important reasons that we explain below.

Since the classification scores depend on the class keys and each participant has access only to the keys of its own classes, it is not possible to use a discriminative loss function, which would require comparing scores across classes. Therefore, we instead formulate a regression-like training objective that aims to maximize the classification score of the true class, without comparing cross-class scores:

$$\max_\theta \sum_{i=1}^{N} \sum_{c \in C^i} \sum_{x \in X^c} \langle \psi(c), \phi_\theta(x) \rangle - \lambda ||\theta||^2, \tag{5}$$

where $N$ is the number of participants, $C^i$ is the set of classes hosted by the participant $i$, $X^c$ is the set of samples belonging to class $c$ and $\lambda$ is the regularization hyper-parameter. In this manner, we aim to learn $\phi_\theta(x)$ such that the resulting embeddings are maximally correlated with the correct class keys.

Here, we emphasize the first reason why it is important to use $\ell_2$-normalized class keys and embedding outputs: in this manner, the resulting classification score is, by definition, restricted to the range $[-1, +1]$, which avoids learning a degenerate embedding function that increases the training objective simply by producing excessively large classification scores (only) for few samples.

In our approach, even if an adversary gets involved during training, it cannot target a particular class without knowing its class key. However, there is still a chance that the adversary may target an arbitrary class by using a randomly generated key as the target key, and, aim to reconstruct the samples belonging to one of the classes without necessarily knowing the identity of the targeted class. In principle, such an attack can be successful if the randomly generated key is sufficiently similar to one of the actual class keys.

To prevent such an attack, it is essential to reduce the probability of approximately replicating class keys by randomly generating keys. Here, one might consider determining and distributing keys in a centralized manner, however, such a prevention technique is not reliable as it is typically not possible to enforce participants to use the assigned keys, *i.e.*, the adversary may still attempt to attack with a random key that it generates privately. Therefore, we need to minimize the probability of generating keys that will lead to scores highly correlated with one of the class scores, without relying on the restrictions on the keys used by the participants, such that an adversary will (most likely) not be successful in training a generative model through a GAN attack.

To address this problem, we rely on high-dimensional, $\ell_2$ normalized class keys such that it becomes unlikely that to generate highly correlated class key through random sampling. In this manner, we are also able to let the participants generate their own keys by sampling each key dimension from a fixed distribution, like $\mathcal{N}(0, 1)$ or $\mathcal{U}(-1,1)$, and $\ell_2$ normalizing the resulting vector. We empirically

---

[2]The final layer of the neural network is an $\ell_2$ normalization layer.

observed that the particular choice of the distribution does not matter as long as we use sufficiently high dimensional vectors, where the resulting vectors end up being approximately orthogonal to each other. We also emphasize that a pair of $\ell_2$ normalized vectors tend to be progressively less correlated as their dimensionality increases (see Section 5.2 for details). By using these nearly orthonormal class keys, even though adversary can still train its local generator, the generator cannot learn data distribution of samples for any class. Hence the GAN attack is prevented, which we empirically demonstrate in Section 5.4.

To further minimize the success probability of the adversary, we propose a modification in model architecture in the next section to safely allow each participant to generate its own class key.

## 4.2 LEARNING WITH HIGH DIMENSIONAL KEYS

In Section 4.1 we state that non-correlated key generation mechanism is required to reduce the probability of the adversary exploiting any classes in the CLF. However, class keys can still be highly overlapping (also discussed in Section 5.2) despite $\ell_2$-normalization, when they are low-dimensional vectors.

In order to reduce the probability of having significant correlation across pairs of randomly and independently generated class keys, we aim to use very high dimensional vectors. However, naively increasing the class key dimensionality undesirably increases the number of trainable parameters, in the last layer. Therefore, this increase in the dimensionality of key vectors also increases the complexity of model architecture which may (i) slow down training significantly, (ii) cause overfitting to training samples, (iii) and therefore, lead to a poor test performance..

To overcome these problems, we propose to add a *fixed* dense layer with an activation function at the last layer of the architecture. By doing so, parameters of this dense layer are stochastically predetermined and kept unchanged throughout training. Thus, the layer does not impose any extra trainable parameters to be learned. It is just used for mapping embeddings to higher dimensions. We let this layer to be shared among all participants. By utilizing a fixed layer, we show that key dimensionality can be increased without effecting convergence of participants so that the GAN attack is prevented as shown in Sections 5.3 and 5.4.

## 5 EXPERIMENTS

In this section we demonstrate that our proposed solution prevents GAN attacks while enabling effective collaborative learning of participants. We first explain the experimental setup in Section 5.1. We discuss the relationship between the key dimensionality and correlation across key pairs (*i.e.* the strength of class keys) in Section 5.2. We verify that our key-based learning formulation performs well in absence of an adversary in Section 5.3. Finally, we empirically demonstrate that our proposed approach prevents privacy against GAN attacks in Section 5.4.

## 5.1 EXPERIMENTAL SETUP

We perform experiments on the well-known MNIST handwritten digits (LeCun et al.) and AT&T Olivetti Faces (Samaria & Harter (1994)) datasets. We choose these datasets for the following reasons:

(i) It is known that GANs are difficult to train (Goodfellow (2017)). Several improvements are proposed for GANs such as Salimans et al. (2016), Tolstikhin et al. (2017), Mirza & Osindero (2014), Che et al. (2016), Warde-Farley & Bengio (2017), Arjovsky et al. (2017), and Arjovsky & Bottou (2017). Therefore, we want the generator to capture statistics of data rather easily. Both of these datasets contain samples that are relatively easy to generate, therefore, these datasets are particularly challenging for our purposes.

(ii) Qualifying the reconstructions obtained by the adversary is relatively easy on these datasets. Thus, we can easily argue about the success of our approach.

We observe that collaborative learning with 5 or more participants is challenging as parameters in the PS overfits quickly to the local dataset of any one of these participants, since local models tend to overfit into their local sets as the number of classes per participant diminishes. An immediate

solution would be to reduce the complexity of local models by reducing the number of learnable parameters. However this affects the generator in a negative way, as the model becomes too simple to extract information about the training examples. As a work around, we carefully tune $\lambda$ on the validation set.

We also found out that learning with fixed layer is also compelling especially when the fixed layer transforms embeddings to a very high dimensional space (*e.g.*, to $\mathbb{R}^{16384}$). We use the following techniques to handle this problem: (i) we use Adam optimizer (Kingma & Ba, 2014) and send the optimizer parameter updates, along with the model updates, to the PS. (ii) we use batch normalization (Ioffe & Szegedy, 2015) (in MNIST) or instance normalization (Ulyanov et al., 2016) (as it behaves better for Olivetti Faces dataset when processing one sample at a time), after the fixed layer. Trainable parameters of batch normalization layer provides extra scale and offset for learning embeddings, which help participants to converge quickly.

For meta-parameters of CLF, in all experiments we set $\theta_d$ and $\theta_u$ to 1.0 as Hitaj et al. (2017) shows that the GAN attack also works for smaller values of $\theta_d$ and $\theta_u$. We also exclude gradient selection mechanism, $\gamma$ and $\tau$. We found out that random and asynchronous orders lead to unstable training of participants, especially when there are 5 or more participants.

We use $\tanh$ activation and $\ell_2$ normalization at the last layer of the local models. We sample class keys from $\mathcal{U}$(-1,1), which empirically behaves well during training. All the experiments are implemented in TensorFlow (Abadi et al., 2015) and the code will be made publicly available upon publication.

## 5.2 KEY SELECTION

In this section we empirically evaluate the importance of key dimensionality in generating strong class keys. For this purpose, we randomly generate 100 $\ell_2$-normalized vectors by sampling from $\mathcal{N}(0, 1)$ or $\mathcal{U}$(-1,1). Then, we find maximum of pair-wise dot products of these vectors. We repeat this process 1000 times and plot the maximum of maximums of the pair-wise dot products in Figure 1. We see that as the size of keys increase, maximum overlap across sampled key pairs diminish.

From this empirical finding, we can claim that it is more robust to generate high dimensional vectors to prevent the GAN attack. In fact, in Section 5.4 we observe that increasing key size weakens the generator, therefore, improves data privacy. However, even when using a fixed layer (instead of a trainable fully connected layer) at the end of the architecture, using excessively large $d_{\text{key}}$ values may lead to instability during neural network training. Therefore, we suggest that $d_{\text{key}}$ should be treated as an hyper-parameter, and, therefore tuned specifically for each model architecture and dataset.

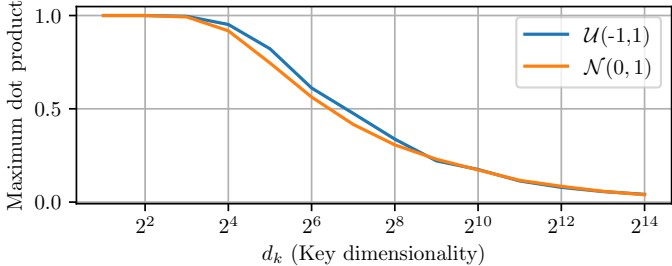

Figure 1: Empirical maximum inner products across pairs of randomly sampled class keys, with respect to key dimensionality. Class key vectors are generated by sampling from from either $\mathcal{N}(0, 1)$ or $\mathcal{U}$(-1,1). X-axis represents size of the vectors and Y-axis shows the maximum of maximum dot product among 100 samples, over 1000 different trials.

## 5.3 COLLABORATIVE LEARNING EVALUATION

In this section, we evaluate our CLF formulation with private class keys, in the absence of an adversary. For this purpose, we examine how the key size $d_{\text{key}}$ affects the test set accuracy of the local

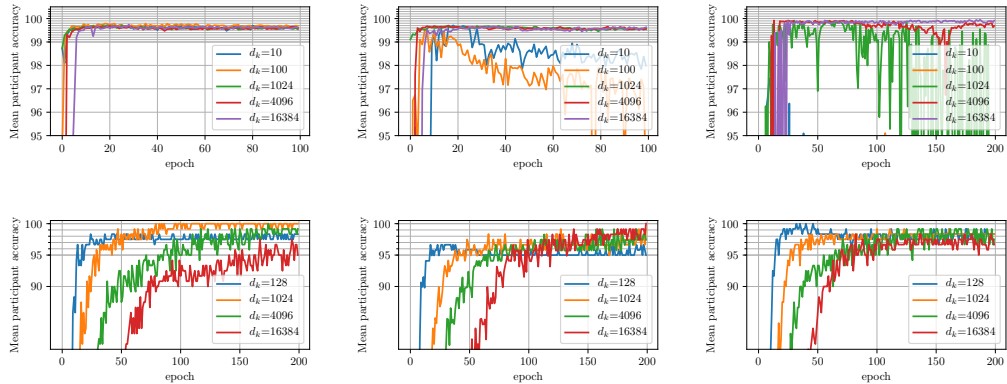

Figure 2: From left to right, mean participant accuracies obtained in collaborative learning over 2, 3, and 5 participants on MNIST (top row) and Olivetti Faces (bottom row) datasets, respectively.

(participant-specific) models, over the training iterations. We present our results in Figure 2, where the top and bottom rows show the results for MNIST and Olivetti Faces datasets, respectively, with 2 (leftmost), 3 and 5 (rightmost) participants.

From the plots for the MNIST dataset, we observe that when there are only 2 participants, the classification accuracy does not vary significantly across the key dimensionalities $d_{key} \in \{10, 100, 1024, 4069, 16384\}$. However, as the number of participants increases (and therefore, the number of classes per participant decreases), key dimensionality starts to become more critical.

Similar observations hold for Olivetti Faces. For this dataset we experiment with 4 different key sizes as $d_{key} \in \{128, 1024, 4069, 16384\}$. Noticeably, we observe a severe tendency towards overfitting (not visible from the figures), which we deal with by using higher regularization hyper-parameter.

## 5.4 PREVENTING GAN ATTACK

In this section we evaluate the success of our approach in preventing GAN attacks. For this purpose, we perform three sets of experiments to cover all aspects of our proposed solution:

  (i) One of the class keys is given to the adversary ($\psi(c_{attack}) : \psi(c)$ for any $c$) when $d_{key}$ is 256 and 128 for MNIST and Olivetti Faces, respectively.
 (ii) Adversary generates random keys that are $\delta$ far (measured in Euclidean distance) from any class key ($\|\psi(c_{attack}) - \psi(c)\| = \delta$ for any $c$) when $d_{key}$ is 256 and 1024 for MNIST and Olivetti Faces, respectively,
(iii) Adversary generates random keys when $d_{key} \in \{128, 1024, 4096, 16384\}$ for MNIST and Olivetti Faces.

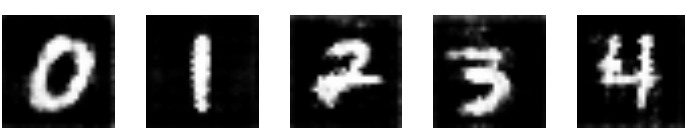

Figure 3: We split MNIST among 2 participants, one being the adversary. The victim has samples of digits 0, 1, 2, 3, and 4. When the exact same key of one of these digits is guessed by the adversary, the local generator of the adversary is able to obtain highly accurate digit reconstructions, which demonstrates the power of GAN attack.

In Experiment (i), we demonstrate the extreme case in which adversary guesses the exact same key of any class in CLF. While this is very unlikely to happen in practice with high-dimensional class keys, we consider this case as a baseline, in which case the GAN attack is expected to be successful.

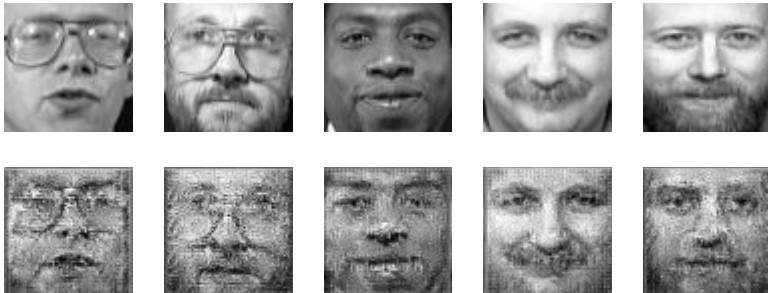

Figure 4: We split Olivetti Faces among 2 participants, one being the adversary. Each participant has face photos of 20 people. We randomly select 5 class labels from the victim, whose private key is available to the adversary. The local generator generates highly successful reconstructions (bottom row) that are highly similar to the original ones (top row), as expected.

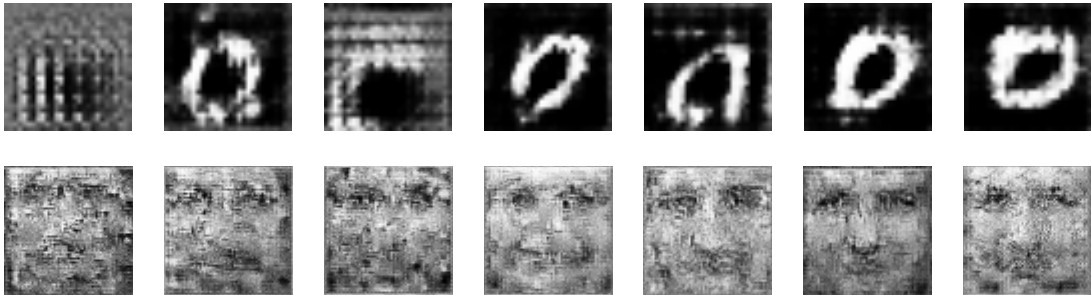

Figure 5: We approximate the maximum Euclidean distance between any class key and $\psi(c_{\text{attack}})$ necessary for the adversary to succeed in attack. From left to right, reconstructions of the adversary when it generates random keys that are $\delta \in \{1.3, 1.2, 1.1, 1.0, 0.5, 0.1, 0.01\}$ far from class key of digit-0 in MNIST (upper row) and person-24 in Olivetti Faces (bottom row), respectively.

The results for this case are presented in Figure 3 and Figure 4. In these results, we observe that the attacker succeeds in reconstructing target class images with high visual similarity, as expected.

In Experiment (ii), we conduct a study to understand the success of the GAN attack as a function of the minimum similarity between the GAN attack key and one of the actual class keys. For this purpose, we provide the attacker a function that takes the key of the victim class ($\psi_{\text{desired}}$), and a predefined degree of similarity ($\delta$), and, generates an $\ell_2$ normalized random key $\psi_{\text{generated}}$ such that the generated key approximates the key of the victim class: $\|\psi_{\text{generated}} - \psi_{\text{desired}}\| \approx \delta$.

We present the results of this experiment in Figure 5. By visually inspecting the reconstructions of the adversary, we deduce that for $\delta$ values larger than $1.2$ in MNIST, and, larger than $0.1$ in Olivetti Faces, the GAN attack produces incomprehensible results.

In Experiment (iii), we show that our model is robust against the GAN attack when there is no constraint on key generation, *i.e.* all keys are randomly and independently generated by the participants, including the guessed ones. We show in Figure 6 and Figure 7 that for sufficiently large key dimensionalities ($d_{\text{key}}$), the GAN attack fails on both MNIST and Olivetti Faces, respectively.

Here we emphasize that, participants may not be able to accumulate a successful model on the PS ,if the last layer in local models is not frozen for $d_{\text{key}} \geq 4096$. As discussed previously, in such cases, local model updates can easily overfit to local datasets. Therefore, the accumulated model ends up being useful for only a few of participants, depending on the order of participants. In addition, the batch-normalization layer with trainable parameters applied after the fixed layer leads to faster convergence as it provides additional shift and scale for the prenormalized embeddings.

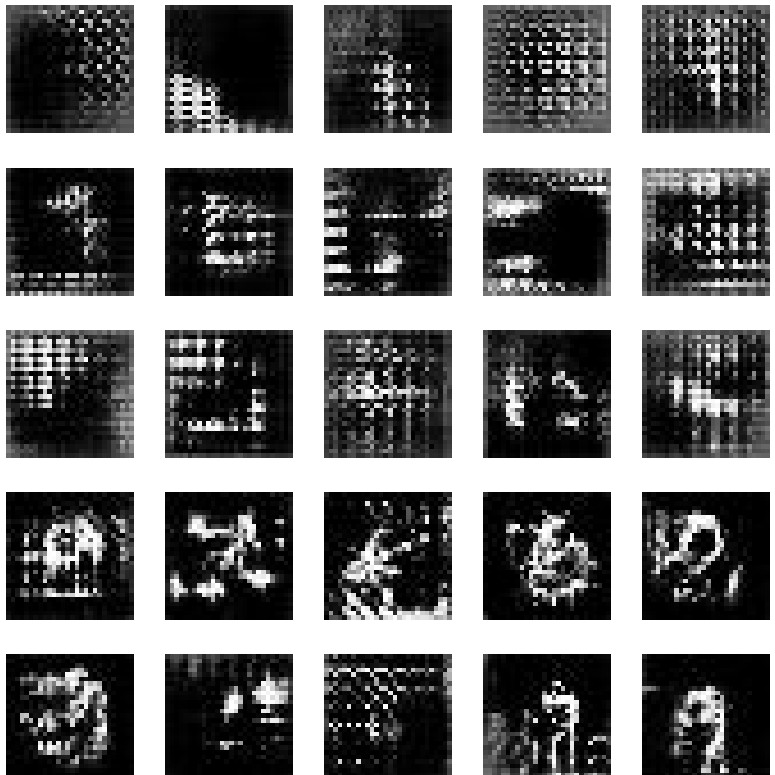

Figure 6: We split MNIST among 5 participants which are also attackers. Each participant creates a random class key, and, trains a local generator with that key. Rows correspond to sample reconstructions obtained by the GAN attack, when $d_{\text{key}}$ is $\{64, 1024, 4096, 16834\}$, respectively. We run the experiments until each participant achieves 97% accuracy on its local dataset. As we increase the key dimensionality, it takes longer for local models to exceed that accuracy threshold thus generators train more. One can see in the last two rows that generators capture a mode, however, the mode is clearly not similar to any of one the MNIST digits. Therefore, the GAN attack fails.

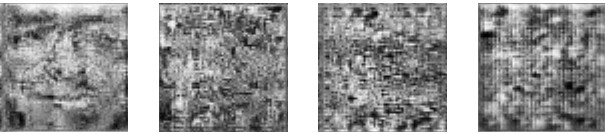

Figure 7: All keys are randomly generated by the participants, including the adversary. From left to right, reconstructions of adversary for $d_{\text{key}} \in \{128, 1024, 4096, 16834\}$, respectively. We show that the mode that GAN learns is likely to belong one of the classes in CLF, for small $d_{\text{key}}$. Yet when $d_{\text{key}}$ is sufficiently large, reconstructions are not realistic faces.

## 5.5 Training with Shared classes across participants

So far, we have assumed that there are no overlapping training classes across the participants. However, as also discussed in Section 4.1, our approach can be utilized even in the cases where participants own samples from shared classes, without sharing their private class keys.

Let $c$ be a class shared by the participants $i$ and $j$. Let there exist two class key vectors in $\mathbb{R}^2$, namely $\psi_i(c)$ and $\psi_j(c)$, which are two distinct keys of class $c$ generated by participants $i$ and $j$, respectively. $\phi_\theta$ is the network that maps input data (*e.g.*, images) into the embedding space. Let $X_i^c$ and $X_j^c$ be the samples that participants $i$ and $j$ have for class $c$. For our analysis, we assume that

the angle between $\psi_i(c)$ and $\psi_j(c)$ are approximately orthogonal, which is correct in practice when the class keys are high-dimensional, $\ell_2$-normalized vectors. In addition, since we expect observing similar examples in $X_i^c$ and $X_j^c$, we assume, for simplicity, that the sets $X_i^c$ and $X_j^c$ are exactly the same, and contain a single sample $x^c$.

Then, since $\|\psi_i(c)\| = \|\psi_j(c)\| = 1$ by definition, and, $\|\phi_\theta(x^c)\| = 1$ due to the $\ell_2$-normalization layer at the output of the network, the training objective, *i.e.* maximization of the dot product between the sample embedding and class key, for $x^c$ can equivalently expressed in terms of minimizing the angle between $\psi_i(c)$ and $\phi_\theta(x^c)$ (denoted by $\alpha_1$), and, between $\psi_j(c)$ and $\phi_\theta(x^c)$ (denoted by $\alpha_2$) [3]. In this simple example devised in $\mathbb{R}^2$, maximizing $\langle\psi_i(c), \phi_\theta(x^c)\rangle$ and $\langle\psi_j(c), \phi_\theta(x^c)\rangle$ with respect to $\theta$ by participants $i$ and $j$ iteratively would converge to a model such that $\langle\psi_i(c), \phi_\theta(x^c)\rangle \approx \langle\psi_j(c), \phi_\theta(x^c)\rangle \approx 0.7$ (*i.e.* $\cos(\pi/4)$), and $\alpha_1 \approx \alpha_2 \approx \pi/4$. We illustrate our claim geometrically in Figure 8a.

In consideration of this behavior in much higher dimensional spaces, we perform the following experiments. For $d_{\text{emb}} = \{64, 256, 1024, 4096, 16384\}$, we generate three $\ell_2$ normalized vectors, namely $\psi_i(c)$, $\psi_j(c)$ and $\phi(x^c)$. Then we update $\phi(x^c)$ to maximize $\langle\psi_i(c), \phi(x^c)\rangle$ and $\langle\psi_j(c), \phi(x^c)\rangle$ until convergence, by simple gradient descent. We repeat this procedure 1000 times and plot maximum of the scores obtained either by $\langle\psi_i(c), \phi(x^c)\rangle$ or by $\langle\psi_j(c), \phi(x^c)\rangle$ for all $d_{\text{key}}$ in Figure 8b. We see that scores converge to 0.7 as we increase the class key dimensionality.

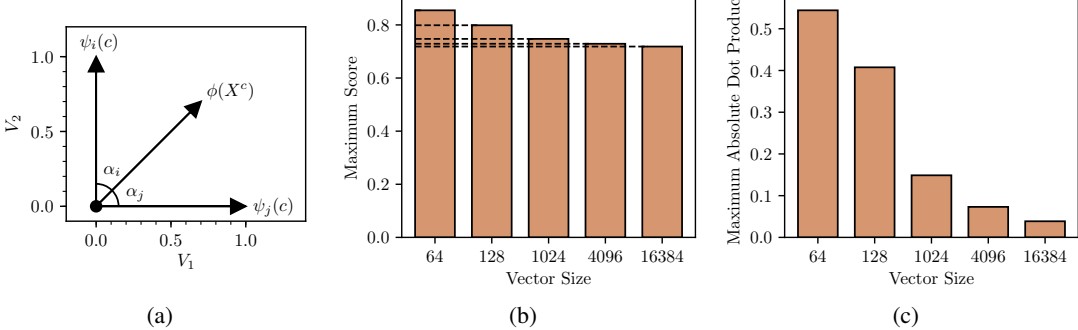

(a)  (b)  (c)

Figure 8: Simulating the case when two participants have samples for a shared class. (a) $\psi_i(c)$ and $\psi_j(c)$ are two nearly orthogonal $\ell_2$ normalized vectors. If we introduce a third random vector, and tune it to maximize its overlapping with $\psi_i(c)$ and $\psi_j(c)$, we would obtain a $\phi(x^c)$ such that $\alpha_1$ and $\alpha_2$ are approximately $\pi/4$ (see text for definitions). (b) In order to verify this behavior in higher dimensions, we randomly generate $\ell_2$ normalized $\psi_i(c), \psi_j(c), \phi(x^c) \in \mathbb{R}^{d_{\text{key}}}$ for $d_{\text{key}} \in \{64, 128, 1024, 4096, 16384\}$. Then we find the optimal $\phi(x^c)$ such that its dot product with $\psi_i(c)$ and $\psi_j(c)$ are maximized. Bars indicate that as we increase the dimensionality, $\psi_i(c)$ and $\psi_j(c)$ are more likely to be orthogonal, therefore $\phi(x^c)$ settles in between them making score of approximately 0.7 with both. (c) We generate new $\ell_2$ normalized vectors in $\mathbb{R}^{d_{\text{key}}}$ for $d_{\text{key}} \in \{64, 128, 1024, 4096, 16384\}$, and check their maximum absolute dot product with the final $\phi(x^c)$. We see that as we increase $d_{\text{key}}$, $\phi(x^c)$ converges towards the plane spanned by $\psi_i(c)$ and $\psi_j(c)$. This confirms that the approach is likely to behave well when training over shared classes, despite using different private keys across the participants.

We continue our analysis by measuring the dot product of the tuned $\phi(x^c)$ with random class keys representing other classes. Our purpose is to interpret algebraically the optimization output of the previous experiment. For each tuned $\phi(x^c)$ we generate 1000 $\ell_2$ normalized vectors, $\psi_{\text{new}}(k)$ for $k = 1, ..., 1000$ and plot maximum absolute dot products of all $\psi_{\text{new}}(k)$ and $\phi(x^c)$, in Figure 8c. Results indicate that when $d_{\text{key}}$ is sufficiently high, multiple participants can declare the same set of labels. For each common label, accumulated model would try to map samples belonging to that class among the participants onto a plane spanned by the embeddings defined for that class among the participants. And that plane would be nearly orthogonal to any other class key. At test time, as

---

[3]Remember that $\langle\psi(c), \phi_\theta(x^c)\rangle = \|\psi(c)\|_2 \cdot \|\phi_\theta(x^c)\|_2 \cdot \cos(\alpha) = \cos(\alpha)$

labels are assigned according to the Equation 4, having multiple keys for a class does not constitute an issue, *i.e.* we can simply assign a test sample to the class whose one of the keys maximizes the classification score.

These observations confirm that the approach is likely to behave well when training over shared classes, despite using different private keys across the participants. In simple terms, the network is likely to map samples to points that are highly correlated with all duplicate keys of their ground-truth classes, and highly uncorrelated with the other ones. In fact, we have empirically verified that training with shared classes perform with no observable issues on both MNIST and Olivetti Faces (example results omitted for brevity).

## 6 CONCLUSIONS

Traditional collaborative learning frameworks (Shokri & Shmatikov, 2015) makes large-scale machine learning possible when data owners do not want to share their datasets due to privacy concerns. However, very recently, such techniques are shown to be prone to powerful GAN attacks (Hitaj et al., 2017). In this work, we have presented a novel collaborative learning technique that is resilient to the GAN attack. More specifically, we have reformulated the training objective of participants by introducing random class keys for each class in the framework. This key-based approach provides effective learning over the participants. Moreover, by utilizing high dimensional keys, class scores of an input is protected against an active adversary that may aim to execute a GAN attack. We have verified the effectiveness of our formulation by empirically showing that (i) the adversary is no longer able to choose which class to exploit, and (ii) generator trained by the adversary cannot capture data distribution well enough to expose any class in the framework.

As future work, we plan to (i) further investigate the performance of key-based approach in supervised classification setting without privacy considerations, (ii) incorporate differential privacy techniques to further reduce the success probability of the attack, (iii) evaluate our work in large-scale datasets.

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
