# OpenReview forum: "Key Protected Classification for GAN Attack Resilient Collaborative Learning"
_ICLR.cc/2018/Conference — Reject_

### Official Review · AnonReviewer3 · 2017-11-27
**The weak assumption on the adversary undermines the usefulness of the protection scheme**

**Rating:** 4
**Confidence:** 4

**Review:**

This paper is a follow-up work to the CCS'2017 paper on the GAN-based attack on collaborative learning system where multiple users contribute their private and sensitive data to joint learning tasks. In order to avoid the potential risk of adversary's mimic based on information flow among distributed users, the authors propose to embed the class label into a multi-dimensional space, such that the joint learning is conducted over the embedding space without knowing the accurate representation of the classes. Under the assumption that the adversary can only generate fake and random class representations, they show their scheme is capable of hiding information from individual samples, especially over image data.

The paper is clearly written and easy to understand. The experiments show interesting results, which are particularly impressive with the face data. However, the reviewer feels the assumption on the adversary is generally too weak, such that slightly smarter adversary could circumvent the protection scheme and remain effective on sample recovery.

Basically, instead of randomly guessing the representations of the classes from other innocent users, the adversary could apply GAN to learn the representation based on the feedback from these users. This can be easily done by including the representations in the embedding space in the parameters in GAN for learning.

This paper could be an interesting work, if the authors address such enhanced attacks from the adversary and present protection results over their existing experimental settings.

---

> ### Author Response · Authors · 2017-12-22
> **We need more explanation for your concerns**
>
> In our approach, we protect participants by hiding class scores from any other participant in CLF. For this purpose, we let participants to create private keys for its local training classes. Please note that private keys are completely randomly distributed, and participants do not share any information about their keys throughout training. (The revised paper, we believe, explains the procedure much more clearly.)
>
> Therefore, we do not see how a GAN attack without a guidance score or feedback signal can be executed to reconstruct the private class keys.
>
> We will be more than happy to discuss if you can elaborate this objection.

---

### Official Review · AnonReviewer2 · 2017-11-28
**Interesting idea to mitigate the GAN attack**

**Rating:** 5
**Confidence:** 2

**Review:**

Collaborative learning has been proposed as a way to learn over federated data while preserving privacy. However collaborative learning has been shown to be suscepti
ble to active attacks in which one of the participants uses a GAN to reveal information about another participant.

This paper proposes a collaborative learning framework (CLF) that mitigates the GAN attack. The framework involves using the neural net to learn a mapping of the inp
ut to a high-dimensional vector and computing the inner product of this vector to a random class-specific key (the final class prediction is the argmax of this inner product). The class-specific key can be chosen randomly by each participant. By choosing sufficiently long random keys, the probability of an attacker guessing the key can be reduced. Experiments on two datasets show that this scheme successfully avoids the GAN attack.

1. Some of the details of key sharing are not clear and would appear to be important for the scheme to work. For example, if participants have instances associated with the same class, then they would need to share the key. This would require a central key distribution scheme which would then allow the attacker to also get access to the key.

2. I would have  liked to see how the method works with an increasing fraction of adversarial participants (I could only see experiments with one adversary). Similarly, I would have liked to see experiments with and without the fixed dense layer to see its contribution to effective learning.

---

> ### Author Response · Authors · 2017-12-22
> **Update**
>
> We address the problem of sharing samples for a common class, in the revised version of the paper. We have added a new section (Section 5.5)  where we discuss and empirically verify that participants may have training examples of overlapping classes without sharing their private keys.
>
> We have also added new attacking results for MNIST showing that there can be multiple attackers in CLF (indeed every participant can be an attacker) in Figure-6. For such cases, the GAN attacks still fail without damaging the learning process. The reconstructions show that generators trained by attackers can capture likelihood of data given the guessed key. However these likelihoods are far from data distributions of the handwritten digits. Which is the expected outcome of our methodology and reflects our success.
>
> Furthermore we speak of how we benefit from the fixed layer in Section 5.4. By using a fixed layer, we are able to control complexity of local models, which is crucial in preventing participants to overfit their local datasets in one epoch of local training.

---

### Official Review · AnonReviewer4 · 2017-12-12
**The paper is unclear and needs more work**

**Rating:** 3
**Confidence:** 4

**Review:**

In this paper, the authors proposed a counter measure to protect collaborative training of DNN against the GAN attack in (Hitaj et al. 2017). The motivation of the paper is clear and so is the literature review. But for me the algorithm is not clearly defined and it is difficult to evaluate how the proposed procedure works. I am not saying that this is not the solution. I am just saying that the paper is not clear enough to say that it is (or it is not). From, my perspective this will make the paper a clear reject.

I think the authors should explain a few things more clearly in order to make the paper foolproof. The first one seems to me the most clear problem with the approach proposed in the paper:

1 $\psi(c)$ defines the mapping from each class to a high dimensional vector that allows protection against the GAN attack. $\psi(c)$ is suppose to be private for each class (or user if each class belong only to one user). This is the key aspect in the paper. But if more than one user have the same class they will need to share this key. Furthermore, at test time, these keys need to be known by everyone, because the output of the neural network needs to be correlated against all keys to see which is the true label. Of course the keys can only be released after the training is completed. But the adversary can also claim to have examples from the class it is trying to attack and hence the legitimate user that generated the key will have to give the attacker the key from the training phase. For example, let assume the legitimate user only has ones from MNIST and declares that it only has one class. The attacker says it has two classes the same one that the legitimate user and some other label. In this case the legitimate user needs to share $\psi(c)$ with the attacker. Of course this sounds “fishy” and might be a way of finding who the attacker is, but there might be many cases in which it makes sense that two or more users shares the same labels and in a big system might be complicated to decide who has access to which key.

2 I do not understand the definition of $\phi(x)$. Is this embedding fixed for each user? Is this embedding the DNN? In Eq. 4 I would assume that $\phi(x)$ is the DNN and that it should be $\phi_\theta(x)$, because otherwise the equation does not make sense. But this is not clearly explained in the paper and Eq 4 makes no sense at all. In a way the solution to the maximization in Eq 4 is Theta=\infty. Also the term $\phi(x)$ is not mentioned in the paper after page 5. My take is that the authors want to maximize the inner product, but then the regularizer should go the other way around.

3 In the paper in page 5 we can read: “Here, we emphasize the first reason why it is important to use l2-normalized class keys and embedding outputs: in this manner, the resulting classification score is by definition restricted to the range [-1; +1],” If I understand correctly the authors are dividing the inner product by ||$\psi(c)|| ||$\phi(x)||. I can see that we can easily divide by ||$\psi(c)||, but I cannot see how we can do dive by ||$\phi(x)||, if this term depends on \theta. If this term does not depend on \theta, then Eq 4 does not make sense.

To summarize, I have the impression that there are many elements in the paper that does not makes sense in the way that they are explained and that the authors need to tell the paper in a way that can be easily understood and replicated. I recommend the authors to run the paper by someone in their circle that could help them rewrite the paper in a way that is more accessible.

---

> ### Author Response · Authors · 2017-12-22
> **Update**
>
> We address the problem of sharing samples for a common class, In the revised version of the paper. We have added a new section (Section 5.5)  where we discuss and empirically verify that participants may have training examples of overlapping classes without sharing their private keys. (Taken partially from our answer to AnonReviwer2.)
>
> Thank you very much for pointing out the ambiguity in the formulation. It has been corrected now.
>
> Since \phi_{\theta}(.) is a deterministic mapping that outputs a vector, we just compute the L2 norm of the output vector, simply as a function of the output vector.

---

### Public Comment · (anonymous) · 2017-12-05
**Differential Privacy**

This paper states (page 2, second paragraph):

However, it has recently been shown that [collaborative learning frameworks (CLFs)] can be vulnerable to not only passive attacks, but also much more powerful active attacks, i.e., training-time attacks, for which the CLF with differential privacy fails to prevent the attack and there is no known prevention technique in general (Hitaj et al., 2017). More specifically, a training participant can construct a generative adversarial network (GAN) (Goodfellow et al., 2014) such that its GAN model learns to reconstruct training examples of one of the other participants over the training iterations.

This is given as the motivation for this work, but this statement is very flawed. Hitaj et al. do not "break" differential privacy. The problem is that they use differential privacy with extremely large parameter values, which yields a meaningless privacy guarantee.

Frank McSherry has posted a detailed critique of the Hitaj et al. paper here:

https://github.com/frankmcsherry/blog/blob/master/posts/2017-10-27.md

---

> ### Public Comment · (anonymous) · 2017-12-08
> **RE: Differential Privacy**
>
> The CCS’17 (Hitaj et al.)  paper mentions several times they don't "break" DP or use DP in any way, but they show that DP is inadequate when epsilon is large (as used and implemented by others) or at the record level. See throughout the paper (https://acmccs.github.io/papers/p603-hitajA.pdf) and the conclusions in particular.
>
> So the blog misses several crucial points and this paper ("Key Protected Classification… “) also provides clear evidence of the privacy risks of CLFs.

---

> > ### Public Comment · (anonymous) · 2017-12-10
> > **Misleading**
> >
> > The Hitaj et al. CCS'17 paper is misleading; only upon close scrutiny does one realize that, when they refer to differential privacy, they mean with crazy parameters.
> >
> > It's analogous to claiming that RSA cryptography is broken and then only on page 3 clarifying that what you really mean is that RSA with 16-bit keys is susceptible to a brute force factoring attack.
> >
> > In particular, the above quote from this submission does not clarify this issue. It says "differential privacy fails to prevent the attack" without providing details. This is on its face false, as the default interpretation is "differential privacy with reasonable parameters."

---

> > > ### Public Comment · (anonymous) · 2017-12-14
> > > **RE: Misleading**
> > >
> > > But that's the point, OTHERS have used RSA with 16-bit keys and Hitaj et al. CCS'17 show this is ill-considered. It's an attack paper, no new scheme is proposed. It is reported that properly set DP will thwart these attacks (but at the cost of utility, see the conclusions).

---

> > > > ### Public Comment · (anonymous) · 2017-12-15
> > > > **"differential privacy fails to prevent the attack"**
> > > >
> > > > The above statement in this paper is false or, at best, misleading. The fact that it is attributed to someone else, doesn't change that.

---

> ### Public Comment · (anonymous) · 2017-12-10
> **related work**
>
> Differential privacy is tangential to the work in this submission and the flaws of the Hitaj et al. paper should not be held against it.
>
> I am commenting because the quote about the related work needs to be clarified. Both Shokri & Smatikov and Hitaj et al. use differential privacy with extremely large parameters, which render it meaningless.

---

> ### Comment · AnonReviewer4 · 2017-12-12
> **Really?**
>
> This reviewer does not have a problem with the paper under study, but believes that Hitaj et al. paper is wrong.
>
> My take is that this review should be removed, because it is only concern with the validity of a already publish work and they should talk to CCS'17 committee about it.
>
> Also, the code for Hitaj et al. 2017 is available if the reviewer thinks the parameters are incorrectly set, they should work with the code to show that the authors maliciously played with the parameters and publish a paper or a blog showing why it does not work. The blog link above does not do that. I think this is the best way to show that Hitaj et al. is not valid. But trashing other conferences with grievances is an old technique that some people use all too frequently and it is becoming really tiring.

---

> > ### Public Comment · (anonymous) · 2017-12-12
> > **Relevance**
> >
> > This submission makes a false statement. It is mathematically impossible to reconstruct training examples while satisfying differential privacy. That statement needs to be corrected. And it is relevant to the motivation for this work.
> >
> > I did not mean to start a debate about the Hitaj et al. paper. My comment is only about the false statement in this submission, which is justified by citing the Hitaj et al. paper.

---

### Author Response · Authors · 2017-12-22
**Clarification on DP**

We thank for the interesting comments and suggestions in this thread. We have just published the comprehensively revised paper where we have removed all of the controversial arguments regarding differential privacy (DP), as suggested by the reviewers.

Our paper, however, is not (directly) about DP: we show that our proposed approach allows privacy-preserving collaborative training without introducing DP or other techniques that corrupt model parameters / parameter updates with noise injection. More importantly, our CLF formulation is resilient against active GAN attacks (Hitaj et.al. 2017).

In more detail, there are two main reasons why we think our approach is of significance:

(1) DP typically requires making a difficult trade-off decision between model accuracy and privacy. In particular, the privacy budget per parameter plots in Shokri et al. (2015) show that in order to reach an acceptable (90%) level of test-set accuracy on MNIST, one may need to use very high "epsilon" values (ie. very low noise), which may significantly reduce the effectiveness of DP in terms of privacy preservation.  Our approach does not necessarily involve such a trade-off between privacy and accuracy (except that using excessively high-dimensional class keys may lead to issues during training).

(2) Our approach prevents CLF against GAN attacks (Hitaj et al. 2017), which can be difficult to avoid using DP, without (significantly) sacrificing the classification accuracy.

Therefore, in summary, what we propose is not built upon DP, instead, it can be seen as a new and alternative approach for privacy preserving collaborative training that builds upon participant-specific keys, as opposed to hiding information through mixing models updates/parameters with noise.

---

### Decision · Program_Chairs · 2018-01-29
**ICLR 2018 Conference Acceptance Decision**

**Decision:**

Reject

**Comment:**

While the reviewers feel there might be some merit to this work,  they find enough ambiguities and inaccuracies that I think this paper would be better served by a resubmission.